# Potential Rheumatoid Arthritis-Associated Interstitial Lung Disease Treatment and Computational Approach for Future Drug Development

**DOI:** 10.3390/ijms25052682

**Published:** 2024-02-26

**Authors:** Eunji Jeong, Hyunseok Hong, Yeon-Ah Lee, Kyoung-Soo Kim

**Affiliations:** 1Department of Medicine, College of Medicine, Kyung Hee University, Seoul 02447, Republic of Korea; eunji0108@khu.ac.kr; 2Yale College, Yale University, New Haven, CT 06520, USA; hyunseok.hong@yale.edu; 3Department of Clinical Pharmacology and Therapeutics, School of Medicine, Kyung Hee University, Seoul 02447, Republic of Korea; 4Division of Rheumatology, Department of Internal Medicine, Kyung Hee University Hospital, Seoul 02447, Republic of Korea; aprildaum@hanmail.net; 5East-West Bone & Joint Disease Research Institute, Kyung Hee University Hospital at Gangdong, Seoul 05278, Republic of Korea

**Keywords:** rheumatoid arthritis-associated interstitial lung disease (RA-ILD), pulmonary fibrosis, anti-fibrotics, computational drug repositioning, drug–target interaction

## Abstract

Rheumatoid arthritis (RA) is a systemic autoimmune disease characterized by swelling in at least one joint. Owing to an overactive immune response, extra-articular manifestations are observed in certain cases, with interstitial lung disease (ILD) being the most common. Rheumatoid arthritis-associated interstitial lung disease (RA-ILD) is characterized by chronic inflammation of the interstitial space, which causes fibrosis and the scarring of lung tissue. Controlling inflammation and pulmonary fibrosis in RA-ILD is important because they are associated with high morbidity and mortality. Pirfenidone and nintedanib are specific drugs against idiopathic pulmonary fibrosis and showed efficacy against RA-ILD in several clinical trials. Immunosuppressants and disease-modifying antirheumatic drugs (DMARDs) with anti-fibrotic effects have also been used to treat RA-ILD. Immunosuppressants moderate the overexpression of cytokines and immune cells to reduce pulmonary damage and slow the progression of fibrosis. DMARDs with mild anti-fibrotic effects target specific fibrotic pathways to regulate fibrogenic cellular activity, extracellular matrix homeostasis, and oxidative stress levels. Therefore, specific medications are required to effectively treat RA-ILD. In this review, the commonly used RA-ILD treatments are discussed based on their molecular mechanisms and clinical trial results. In addition, a computational approach is proposed to develop specific drugs for RA-ILD.

## 1. Introduction

Rheumatoid arthritis (RA) is an autoimmune disease with a systemic inflammatory response characterized by joint inflammation and swelling. Overactive immune systems caused by RA commonly result in extra-articular manifestations, such as cardiovascular and respiratory dysfunction [1]. Approximately 66% of patients with RA have lung involvement in the form of interstitial, airway, pleural, vascular, and parenchymal diseases [2,3]. The most prevalent form of lung involvement in patients with RA is interstitial lung disease (ILD), which is the chronic inflammation of the interstitial space that causes fibrosis and scarring of the lung tissue [4]. Rheumatoid arthritis-associated interstitial lung disease (RA-ILD) constitutes 13% of the total deaths of patients with RA, and patients with RA-ILD show a significantly low 5-year survival rate of 35–39% [5]. Overall, patients with RA-ILD have a death risk three times higher than those with RA without ILD [6].

Pulmonary fibrosis is the fundamental cause of the high mortality and morbidity rates in patients with RA-ILD. The progression of fibrosis can be observed in wide fibrotic areas in the lungs using computed tomography scans with histological patterns, such as usual or nonspecific interstitial pneumonia (abbreviated as UIP and NSIP, respectively) [7]. Aggravated lung function caused by fibrosis results in a decline in forced vital capacity (FVC) and gas exchange (diffusing capacity of the lungs for carbon monoxide [DLCO]) while reducing exercise capacity and the health-related quality of life of patients [1]. If the condition worsens during successive fibrotic events, acute exacerbation, infection, or malignancy, death can occur.

The control of fibrosis is important for determining the prognosis of RA-ILD. However, predicting and diagnosing the onset of fibrotic phases in RA-ILD remain challenging. Genetic factors, such as the rs35705950 variant in the MUC5B gene and short telomere length, or environmental factors, such as smoking, sex, age, underlying diseases, and a high rheumatoid factor, increase the risk of ILD; however, a distinct correlation remains lacking [8,9]. Although genetic biomarkers, such as KL-6, SP-D, and HLA-DR2 alleles, or antibodies in the serum of patients with RA, such as anti-cyclic citrullinated or anti-citrullinated protein antibodies, were suggested as biomarkers to detect the onset of fibrosis, their detection accuracy is limited [10]. Owing to difficulties in identifying the emergence of fibrosis, the rudimentary inhibition of immune responses and hindrance of fibrosis progression are of vital importance in RA-ILD treatment.

Currently, the first-line treatment for RA-ILD primarily relies on drugs used for idiopathic pulmonary fibrosis (IPF) because the two diseases share similar clinical, radiological, and molecular patterns. Two major drugs, namely, pirfenidone and nintedanib, are FDA-approved anti-fibrotic agents for IPF that are commonly used as treatments for RA-ILD. In the case of nintedanib, the treatment recently received FDA approval as the first treatment for chronic fibrosing ILD with a progressive trait, supporting its use in RA-ILD. These anti-fibrotic drugs can directly interfere with specific signaling pathways to block or slow the progression of fibrosis and reverse the morphological changes induced by fibrosis. Several immunosuppressants and disease-modifying antirheumatic drugs (DMARDs), which are the primary treatments for RA, were also shown to slow fibrotic progression in RA-ILD by reducing the autoimmune response, which is the causative factor of pulmonary fibrosis. 

However, the limitations of current drugs and the need for basic inhibition of the autoimmune-induced fibrosis phase necessitate the development of new drugs for RA-ILD. While continuous research and development is in progress, drug repositioning is one of the primary strategies to be explored. Because multi-organ fibrotic diseases share similar clinical and pathological characteristics, these drugs can be repurposed to benefit the fibrotic manifestation of RA-ILD. With adequate targets that can hinder the progression of fibrosis in RA-ILD and a set of drug candidates, computational analysis can aid and accelerate the process by predicting the drug–target interactions (DTIs) between them to recommend favorable pairs for further confirmation.

## 2. Anti-Fibrotic Agents for ILD

To date, the most widely used drugs for the treatment of RA-ILD are pirfenidone and nintedanib, which were initially developed and approved by the Food and Drug Administration (FDA) for IPF. Their anti-fibrotic and anti-inflammatory potentials stem from the strong resemblance of RA-ILD and IPF in terms of pulmonary fibrotic patterns and prognosis. RA-UIP, which is a prevalent subtype of RA-ILD, is considered the most analogous to IPF because both show identical UIP fibrotic patterns and similar prognoses. RA-ILD and IPF are associated with pulmonary fibrosis and acute exacerbations, respectively. In addition, the upregulation of several genetic markers, such as eotaxin, IL-8, MMP-2, MMP-7, MMP-9, and Flt-3L, and genetic mutations in TERT, RTEL1, PARN, and SFTPC were observed in patients with RA-ILD and IPF [11,12]. These similarities support the implementation of therapeutic strategies for IPF in patients with RA-ILD.

However, RA-ILD and IPF differ in several aspects that can cause adverse effects during treatment. RA-ILD is a multi-system disease that stems from an autoimmune response. In contrast, IPF includes only lung-limited fibrosis without a specific cause. Therefore, their responses to treatment may differ based on their contrasting pathogenic pathways. For instance, an immunosuppressive approach to IPF therapy is contraindicated due to its inefficiency and increased mortality and morbidity [13]. However, immunosuppression is the standard of care for patients with RA-ILD because the disease is derived from an autoimmune disorder [14]. Moreover, while IPF exclusively presents UIP histological patterns in pulmonary fibrosis, RA-ILD can include other patterns, such as NSIP, organizing pneumonia (OP), and mixed or indeterminate forms [15]. Different histological patterns affect the disease prognosis and treatment effectiveness. RA-ILD with NSIP or OP responds better than RA-UIP to immunosuppressive treatment while also showing a preferable disease prognosis [16,17]. Thus, while the similarities between RA-ILD and IPF can guide the discovery of novel treatments for RA-ILD, their differences should be considered in the process of drug development. This section provides an overview of the anti-fibrotic mechanisms and clinical trials of pirfenidone and nintedanib in patients with RA-ILD.

### 2.1. Nintedanib

Nintedanib is an FDA-approved anti-fibrotic agent used for the treatments of IPF, systemic sclerosis-ILD, and chronic fibrosing ILD with a progressive phenotype, regardless of the underlying cause [18]. Using nintedanib for the treatment of RA-ILD has attracted increased attention. Nintedanib functions as a tyrosine kinase inhibitor (TKI) for several receptors that contribute to fibrogenesis, including PDGFR α and β, FGFR 1–3, vascular endothelial growth factor receptor (VEGFR) 1–3, colony-stimulating factor 1 receptor (CSF1R), and lymphocyte-specific tyrosine protein kinase (Lck) [19]. Thus, it exerts its anti-fibrotic effect via a multifaceted inhibitory mechanism. It also regulates the activity of Lck, which is essential for T cell proliferation and the release of pro-fibrotic mediators, such as IL-2, IL-4, IL-5, IL-10, IL-12p70, IL-13, and interferon gamma (IFN-γ) [20,21]. By specifically targeting PDGF, FGF2, VEGF-A, and other growth factors, nintedanib suppresses the function of fibrocytes, consequently inhibiting fibrocyte migration, differentiation, and proliferation [22]. The study also mentioned the ability of nintedanib to suppress transforming growth factor-beta (TGF-β)-induced fibroblast–myofibroblast transition and TGF-β-stimulated collagen secretion [22]. In pulmonary fibrosis, nintedanib significantly decreases certain markers that indicate M2 macrophage polarization. M2 macrophage polarization is a major contributor to pulmonary fibrogenesis because it releases fibrotic factors, such as TGF-β and gelectin-3, which accordingly stimulate the epithelial–mesenchymal transition (EMT), endothelial–mesenchymal transition (EndoMT), and fibroblast proliferation [23,24].

Several clinical trials were conducted using nintedanib in patients with RA-ILD. The INBUILD trial was a phase 3 clinical trial conducted to evaluate the efficacy of nintedanib in patients with progressive fibrosing ILD [25]. Of the 663 patients enrolled in the trial, 89 had RA-ILD, and a retrospective subgroup analysis was performed [26]. Compared with the placebo, nintedanib demonstrated an improvement in patients with RA-ILD by significantly lowering the average rate of FVC decline by 59% over 52 weeks and reducing the risk of acute exacerbation or death. The most common adverse effect associated with nintedanib was diarrhea, which occurred in 61.9% of patients in the nintedanib group; however, the symptoms were still manageable. In addition, its concomitant use with other DMARDs showed no differences in effectiveness or safety profile. Thus, nintedanib demonstrated sufficient efficacy and safety as a potential treatment option for patients with RA-ILD. A recent retrospective cohort study from Portugal suggested the importance of the early initiation of nintedanib treatment in halting ILD progression. Patients treated with nintedanib within 3 years of their initial ILD diagnosis exhibited significantly improved respiratory function within 24 months of treatment [27]. 

Nintedanib also affects arthritis in murine models. In a zymosan-induced arthritis SKG mouse model, nintedanib treatment reversed joint swelling and decreased the arthritis score compared with a placebo saline treatment. Moreover, while the talus and calcaneus bones of the nintedanib-treated mice were negligibly changed, those of the saline-treated mice were irregular and deformed, suggesting severe erosion [28]. Based on these in vivo results, future clinical trials are required to determine the efficacy of nintedanib against arthritis involvement.

### 2.2. Pirfenidone 

Pirfenidone has been FDA-approved for IPF treatment since 2014. Phase 3 trials, such as ASCEND in 2014 and CAPACITY 1 and 2, supported its efficacy and safety for IPF treatment as an anti-fibrotic agent. Thus, several attempts have been made to use pirfenidone to treat diseases similar to pulmonary fibrosis, including RA-ILD. 

Pirfenidone is a pyridine derivative with anti-fibrotic, anti-inflammatory, and anti-oxidant properties, which collectively contribute to the inhibition of pulmonary fibrosis. The anti-fibrotic activity of pirfenidone originates from suppressing TGF-β1, which is a master regulator of fibrosis. Through TGF-β1, pirfenidone reduces fibroblast proliferation, myofibroblast differentiation, extracellular matrix (ECM) production, and TGF-β1-induced SMAD3 phosphorylation [29]. Anti-inflammatory effects of pirfenidone can be observed via a translational level decrease of inflammatory cytokines, such as TNF-α, IL-6, and IFN-γ, and the upregulation of anti-inflammatory cytokines, such as IL-10 [30]. By controlling the balance between pro-inflammatory cytokines, pirfenidone regulates tissue injury and the subsequent restoration process of wound healing. Regulating oxidative stress is also crucial for pulmonary fibrosis treatment because reactive oxygen species participate in the diverse molecular mechanisms of fibrogenesis [31]. By eliminating hydroxyl radicals and controlling oxidative stress, pirfenidone establishes itself as a potent anti-fibrotic agent [32].

In addition to its strong anti-fibrotic capacity, pirfenidone has been repurposed as an antirheumatic drug. Gan et al. [33] demonstrated a reduction in joint synovial inflammation and swelling in a collagen-induced arthritis rat model treated with pirfenidone. In vitro and bioinformatic analyses supported the antirheumatic effect of pirfenidone via the inhibition of inflammatory cytokines and cell migration, rendering it more suitable for RA-ILD treatment.

Multiple clinical trials demonstrated the promising potential of pirfenidone for the treatment of RA-ILD. Recently, a phase 2 trial (TRAIL-1) using pirfenidone in patients with RA-ILD was attempted but was terminated early due to the COVID-19 pandemic. However, certain data indicated that pirfenidone was effective in the treatment of RA-ILD. Pirfenidone lowered the rate of FVC decline compared with the placebo without presenting any new adverse effects. In addition, this study demonstrated that pirfenidone combined with DMARD therapy had an acceptable safety profile and tolerability [34]. The RELIEF trial, which was another phase 2 study with 127 non-IPF patients with ILD, included 17 patients with RA-ILD. In the trial, patients treated with pirfenidone experienced a lower decline in FVC, indicating decelerated pulmonary aggravation, compared with those who received the placebo, showing that pirfenidone alleviated RA-ILD symptoms [35]. 

A prospective controlled cohort study was conducted to investigate the efficacy and safety of pirfenidone combined with immunosuppressant therapy in patients with connective-tissue-disease-associated interstitial lung disease (CTD-ILD). Of the 111 patients, 17 were diagnosed with RA-ILD. While all patients in the study were prescribed immunosuppressants, those in the test group concomitantly received pirfenidone. Pirfenidone-treated patients with RA-ILD exhibited an enhancement in DLCO%, which is a measure of gas exchange, by 7.40% compared with the baseline, while the control group showed a 5.5% decrease [36].

A phase 2 study of pirfenidone for the treatment of unclassifiable progressive fibrosing ILD was also conducted. Although patients diagnosed with RA-ILD were not included in the study population, pirfenidone enhanced pulmonary function and exercise capacity compared with a placebo without presenting extensive adverse effects [37]. When an ongoing randomized controlled trial for patients with CTD-ILD with a subpopulation of patients with RA-ILD is completed as expected, we will obtain a clearer picture of the potential of pirfenidone for RA-ILD treatment [38].

## 3. Conventional Synthetic DMARDs for RA-ILD

Immunosuppressants are early-stage drugs used to treat patients with RA. Immunosuppressive agents can hinder the progression of RA to RA-ILD by reducing the autoimmune response that is responsible for lung fibrosis (Figure 1). Although there are currently no FDA-approved immunosuppressants for improving pulmonary function in patients with RA-ILD, several drugs that have shown efficacy against similar or related diseases are being tested. Thus, several immunosuppressants should be considered as plausible treatments for RA-ILD.

### 3.1. Methotrexate (MTX)

MTX, which is a folate derivative, is considered the first-line DMARD for RA. However, its use for RA-ILD treatment is limited owing to MTX-related pulmonary adverse effects, including hypersensitivity pneumonitis, that are presumed to be associated with RA-ILD progression [39]. Several studies have since overturned the original beliefs of the clinicians. In a retrospective multicenter study conducted by Kiely et al. [40], MTX exposure did not result in RA-ILD but rather delayed its onset. Juge et al. [41] further advocated for the positive effect of MTX using a case–control study that revealed a lower risk of RA-ILD in MTX ever-users compared with MTX never-users, with an average delay of 3.6 years in ILD detection. MTX was also suggested to be a protective factor against lung function decline and mortality in cohorts of patients with RA-ILD when provided after RA-ILD incidence [42]. A phase 4 randomized controlled trial (PULMORA) is in progress to compare the effects of MTX with those of the Janus kinase (JAK) inhibitor tofacitinib [43]. Through this clinical trial, we expect to gain a better understanding of the effects of MTX on RA-ILD.

### 3.2. Cyclophosphamide (Cyc)

Cyc is a DNA alkylating immunosuppressive agent that hydrolyzes into phosphor mustard when activated, consequently resulting in DNA damage and the inhibition of cell proliferation [44]. Although the underlying mechanism is not fully understood, the immunosuppressive effect of Cyc was shown to benefit patients with RA-ILD [45]. A 10-year retrospective study in Germany by Schupp et al. [46] demonstrated the ability of Cyc to stabilize lung function in patients with both inflammatory and fibrotic ILD. Randomized controlled trials also support the ability of Cyc to improve lung conditions in patients with CTD-ILD with better FVC results compared with a placebo [47,48]. Cyc also relieved tender and swollen joints in patients with RA, which confirms its antirheumatic effects [49]. However, certain studies have reported that the mortality rate of patients with acute exacerbation of RA-ILD treated with Cyc did not significantly differ from that of the control group [50]. Moreover, the administration of Cyc alone is not recommended because of its high toxicity and adverse effects [4].

### 3.3. Mycophenolate Mofetil (MMF)

MMF is less toxic than Cyc and inhibits inosine-5′-monophosphate dehydrogenase and de novo guanine syntheses, which are critical elements for DNA synthesis and the proliferation of T and B lymphocytes [51]. In addition to its immunosuppressive effects, studies supported the anti-fibrotic capability of MMF by suppressing fibroblast proliferation and the expression of fibrosing cytokines, such as TGF-β and IL-6 [52,53]. Because MMF can control both the inflammatory response and pulmonary fibrosis, it is considered a promising option for the treatment of RA-ILD. Although controlled studies of MMF for RA-ILD are still needed, MMF has shown positive results in several studies, with improved or stabilized pulmonary physiology in patients with CTD-ILD, including subgroups of patients with RA-ILD [54,55]. While the frequent adverse events of MMF include gastrointestinal symptoms, bone marrow suppression, and infection due to immunosuppression, it remains an appealing option compared with the safety profiles of other immunosuppressive drugs [56]. However, MMF is known to have no benefit in treating the articular manifestations of RA [57]. Thus, it is recommended to be used in combination with a different immunosuppressant that can target joint involvement.

### 3.4. Azathioprine (AZA)

AZA is an antimetabolite that acts as an immunosuppressive agent through the inhibition of purine synthesis and DNA replication of lymphocytes [58]. AZA already demonstrated its advantage on pulmonary function for similar ILDs, such as systemic-sclerosis-associated interstitial lung disease (SSc-ILD), sarcoidosis, and chronic hypersensitivity pneumonitis [59,60,61]. Therefore, AZA is commonly used along with other immunosuppressants for early RA-ILD treatment. A multicenter retrospective cohort study performed by Matson et al. [62] supported the notion of using AZA for RA-ILD. A total of 212 patients with RA-ILD on immunosuppressive therapy were enrolled in the study, including 92 patients who were treated with AZA. Regardless of the type of immunosuppressive agents, patients treated with immunosuppression exhibited a 3.9% increase in FVC and a 4.5% increase in DLCO compared with the predicted value if the pretreatments had continued. Also, a retrospective study of 56 patients with CTD-ILD treated with AZA concluded that AZA can stabilize or improve CTD-ILD as long as the patient can tolerate the drug [63]. However, there are also concerns about AZA’s safety profile. PANTHER-IPF trial was a randomized, double-blind, placebo-controlled trial for IPF with concomitant treatment using prednisone, AZA, and N-acetylcysteine, but it was terminated early due to a high mortality rate [13]. Oldham et al. also noted that while AZA and MMF exhibited comparable efficacy on pulmonary stability, AZA had a higher rate of adverse effects and drug intolerance when treating CTD-ILD. [64]. Therefore, the prescription of AZA to patients, especially with RA-UIP, which is a subtype of RA-ILD similar to IPF, should be done with caution.

### 3.5. Tacrolimus (TAC)

Tacrolimus is a type of calcineurin inhibitor that inhibits calcium-dependent events, including T lymphocyte signal transduction and IL-2 transcription [65]. There have been several attempts to treat ILD using TAC. For instance, a retrospective study was conducted with 49 patients with ILD associated with polymyositis or dermatomyositis [66]. In the study, the patients treated with TAC along with conventional therapy showed significantly longer event-free and disease-free survival compared with the group only treated with conventional therapy, suggesting the potential of TAC to improve the prognosis of ILD. Similar research was done with antisynthetase-associated ILD. A cohort study of patients with antisynthetase-associated ILD revealed that 13 out of 15 patients who received a calcineurin inhibitor showed stable or improved FVC [67]. To gain a better understanding of TAC’s efficacy against CTD-ILD, Yamano et al. [68] conducted a retrospective study of patients with CTD-ILD who received combination therapy of TAC and corticosteroid. Among the study population of 28 patients, 11 patients were diagnosed with RA-ILD. After 1 year of TAC therapy, patients showed significantly improved lung physiology and exercise capacity without severe adverse effects. Recently, a Korean RA-ILD cohort also revealed that tacrolimus delayed the progression of RA-ILD compared with a non-treated group, but with non-statistical significance [69].

## 4. Biologic DMARDs

Antibodies can disable specific cells and the immune system, primarily during organ rejection or in autoimmune diseases (Figure 1). Rituximab (RTX) is an anti-B-cell antibody widely used to alleviate joint inflammation in RA. Recently, clinical trials showed their auxiliary efficacy, not only in joint inflammation but also in delaying fibrosis. Anti-tumor necrosis factor alpha (TNF-α) agents have limited clinical evidence against RA-ILD, but are widely used in RA treatments. 

### 4.1. Rituximab (RTX)

RTX is a chimeric monoclonal antibody that targets the CD20 surface marker on both B cell precursors and mature B cells [70]. RTX binding can result in B cell depletion via several mechanisms, including complement-dependent cytotoxicity, cell-mediated cytotoxicity, and apoptosis [71]. With its anti-arthritic effect, RTX treatment showed clinically significant improvements with articular involvement and was approved by the FDA as a biologic DMARD for RA in the REFLEX and DANCER trials [72,73,74]. Promising results were also observed in the lung function of patients with CTD-ILD and RA-ILD, showing improved or maintained pulmonary function and slower progression of the disease [75,76,77]. Recently, a randomized controlled trial was conducted to compare the efficacy of RTX and Cyc in patients with CTD-ILD (RECITAL). This study demonstrated the ability of RTX and Cyc to improve the pulmonary function and quality of life of patients without significant differences, with RTX causing fewer adverse effects [48].

### 4.2. TNF Inhibitors

Although few clinical studies have been conducted, three TNF-α inhibitors, namely, infliximab, adalimumab, and etanercept, exhibit positive effects on patients with RA-ILD. The intravenous infusion of infliximab showed the radiographic stabilization of pulmonary functions, such as dyspnea, cough, and exercise tolerance, and improvement in joints [78]. However, a retrospective study of 100 patients with RA-ILD demonstrated that the anti-TNF-α therapy of adalimumab and infliximab should be used with caution in older patients with RA-ILD because lung complications can occur within months of initial anti-TNF-α treatment [79]. Etanercept was also suggested to be used with caution because the exacerbation of ILD was observed in a specific case [80].

### 4.3. Abatacept (ABA)

ABA is a co-stimulation blocker comprising the extracellular domain of human cytotoxic T-lymphocyte-associated antigen 4 (CTLA4) and a modified Fc portion of human immunoglobulin G1. By binding to CD80/CD86, ABA hinders the CD80/86-CD28 interaction, consequently inhibiting co-stimulatory signals essential for T cell activation [81]. Although ABA is an FDA-approved drug for the treatment of RA, studies showed that it also has favorable effects on the treatment of RA-ILD. In vivo studies demonstrated the potency of ABA in inhibiting lung inflammation and fibrosis in murine models [82,83]. In an observational study of 263 patients with RA-ILD, approximately 80% of patients treated with ABA maintained stable or improved pulmonary function with better FVC and DLCO, as well as a reduced DAS28esr score, indicating RA disease activity [84]. Other studies and systematic reviews also supported the efficacy and safety of ABA in patients with RA-ILD [85,86,87]. Although it was terminated because of the COVID-19 pandemic, a phase II study of ABA in patients with RA-ILD (APRIL) indicated an acceptable safety profile with no serious adverse reactions [88]. Thus, randomized controlled trials are necessary to further investigate the potential of ABA as a treatment for RA-ILD.

### 4.4. Tocilizumab (TCZ)

Tocilizumab is an FDA-approved IL-6 inhibitor for SSc-ILD treatment based on the focuSSced and faSScinate trials [89,90]. As SSc-ILD and RA-ILD share similarities as members of the CTD-ILD family, tocilizumab is expected to demonstrate similar efficacy toward RA-ILD. Several studies strengthened this idea with positive results. In a national multicenter retrospective study from Italy, 19 out of 28 with RA-ILD showed improved or stabilized FVC and DLCO after TCZ therapy without significant adverse effects [91]. Furthermore, in the study conducted by Otsuji et al. [92], 34 patients with RA-ILD were treated with TCZ for 6 months to evaluate the safety and efficacy of the treatment. Patients administered with TCZ displayed a significant decrease in both MMP-3 and KL-6, indicating the efficacy of the drug toward both RA and ILD. Patients’ chest CTs also indicated the attenuation of ILD progression after TCZ treatment.

## 5. Corticosteroid (Glucocorticoid)

Glucocorticoids, also known as corticosteroids or steroids, are conventional treatments for autoimmune diseases. They function via multiple mechanisms, including blocking the synthesis of cytokines and cell surface molecules necessary for the activation of the immune system via the inhibition of NF-κB [93]. Owing to their anti-inflammatory and immunosuppressive effects, glucocorticoids are favored for treating inflammatory RA-ILD with NSIP or OP [94,95]. Positive results were also observed with steroid treatments for fibrotic RA-ILD with UIP. In a retrospective study conducted by Song et al. [96], approximately 50% of patients with RA-UIP showed improvement or stabilization following glucocorticoid treatment. In contrast, glucocorticoid treatment in patients with IPF resulted in higher morbidity and mortality; therefore, it is contraindicated in IPF [97,98]. Despite the high histopathological similarity between IPF and RA-ILD, the same treatment might have disputed pharmacological effects. Moreover, the use of glucocorticoids in patients with RA-ILD carries potential risks, such as an increased risk of infection or osteoporosis development; therefore, additional caution is needed [99,100].

## 6. Targeted Synthetic DMARDs (JAK Inhibitors)

JAKs are a group of intracellular tyrosine kinases that participate in signaling pathways that regulate several physiological cell processes, including differentiation and metabolism [101]. Transcription factors, called STATS, which is an oncoprotein family encoded by tumor-associated genes, jointly form a signaling pathway with JAK. JAK and STATS have several subtypes, but JAK2/STAT3 is the most prevalent in ILD [10,102,103]. STAT3 transduces peptide hormone signals from the cell surface to the nucleus and is activated by cytokines, growth factors, and several peptide hormones. While RA-ILD with UIP shares almost identical clinical patterns with IPF, RNA transcriptomic data show that RA-UIP has a nuclear form of phosphorylated JAK2, whereas most patients with IPF have a cytoplasmic form of phosphorylated JAK2 [104]. Such variations imply that applying therapeutic approaches for IPF in patients with RA-ILD may not yield the same effects and may even cause unexpected adverse effects. The JAK/STAT pathway is activated by several pro-fibrotic or pro-inflammatory cytokines, such as IL-6, IL-11, and IL-13, or growth factors, such as platelet-derived growth factors (PDGFs), TGF-β1, and fibroblast growth factors (FGFs) [102]. Blocking these signals inactivates the JAK/STAT pathway, which consequently arrests fibrosis. JAK inhibitors also exhibit anti-fibrotic effects by directly affecting fibroblast activation [105].

Tofacitinib, ruxolitinib, baricitinib, and upadacitinib are FDA-approved JAK inhibitors used for the treatment of RA [106,107,108,109]. They show significant improvement in joints by controlling the plasma levels of pro-inflammatory cytokines, reducing joint destruction, and exerting anti-fibrotic effects by directly affecting fibroblasts. Research on macrophages has revealed that they can be effective in controlling diseases characterized by concomitant inflammation and fibrotic manifestations, which also supports their efficacy against RA-ILD [96,97]. Figure 2 summarizes the treatment options of RA-ILD we have reviewed so far, including JAK inhibitors.

### 6.1. Tofacitinib

Tofacitinib is a JAK 3/2 inhibitor with strong regulatory effects on IL-17A. IL-17A is secreted by T helper 17 cells and functions cooperatively with IL-17RA and IL-17RC during fibrosis. Large areas of IL-17RA-induced fibroblast accumulation and fibrosis were observed in patients with RA-ILD. IL-17A plays a distinct role in RA-ILD compared with other normal lung diseases or even IPF [110]. The suppression of IL-17A ameliorates the activity of inflammatory T helper 17 cells, thereby alleviating fibrosis. In vivo research using SKG mice showed that tofacitinib slowed the progression of RA-ILD by increasing the number of myeloid-derived suppressor cells [111]. A phase 3 randomized controlled clinical trial indicated that 5 and 10 mg of tofacitinib administered twice a day demonstrated maximal healing effects at 6 months. Signs, symptoms, and pain caused by RA were reduced, physical lung function improved, and the progression of fibrosis or structural damage decreased. However, 28% of the patients discontinued the treatment because of adverse events [112]. Tofacitinib can be applied either alone or in combination with immunosuppressants, such as MTX, without affecting its efficacy [113]. For instance, tofacitinib was administered to a 52-year-old man after signs of ILD were diagnosed without immunosuppressants. Tofacitinib successfully controlled frequent inflammation and improved respiratory functions [114]. Multiple studies showed that tofacitinib retains or slows the progression of RA-ILD, reduces the risk of RA leading to ILD, and exceeds other biological DMARDs (bDMARDs) in restoring pulmonary function. The administration of tofacitinib over an 8–12 month period to two patients with RA-ILD successfully controlled inflammation in the joints without exacerbations or hospitalizations due to lung disease [115]. In another study, among 31 patients with RA-ILD, 13 were treated with tofacitinib, and 83.9% exhibited overall stability or improvement in RA-ILD symptoms [116]. Tofacitinib exceeded other bDMARDS in reducing the risk of ILD, and another post hoc analysis of 21 clinical trials supported this finding by concluding that the incidence rate of ILD following tofacitinib treatment was extremely low (0.18).

### 6.2. Ruxolitinib

Ruxolitinib, which is a specific JAK2 inhibitor, regulates the activity of pro-fibrotic M2 and pro-inflammatory M1 macrophages [117]. IL-13- and IFN-γ-activated macrophages followed a signaling pathway that involved JAK2, to which ruxolitinib indicated direct repression. Polarization markers, such as MHC II and Toll-like receptor 4 (TLR4), which indicate the differentiation processes of M1 and M2, respectively, were decreased, as well as the secretion of pro-inflammatory cytokines, such as CXCL10, IL-6, and TNF-α [117].

Although clinical trials of ruxolitinib for patients with RA-ILD are absent, few case reports support ruxolitinib’s anti-fibrotic capacity for ILD. In the case report by Manuel et al. [118], ruxolitinib administration improved the pulmonary function of two patients with ILD from STAT3 gain-of-function mutation without major adverse events. The administration of ruxolitinib showed similar CT scan changes and improvement in clinical symptoms, while the FEV rates significantly increased. Another case study suggested ruxolitinib’s efficacy and safety in treating a patient with refractory systemic idiopathic juvenile-arthritis-associated ILD [119]. Ruxolitinib was well tolerated and when concomitantly administered with corticosteroid, the corticosteroid administration dosage tapered.

### 6.3. Baricitinib

Baricitinib is a JAK1/JAK2 inhibitor suggested as a treatment for patients with RA-ILD who do not respond to the initial treatment with MTX [120]. An analysis of 3770 patients with RA from eight randomized clinical trials and one long-term extension study on baricitinib showed promising effects in lowering the risk of ILD development [121]. When 11 patients with RA and 4 patients with RA-ILD were treated with baricitinib for 6 months, a considerable decrease in inflammation was observed in patients with RA-ILD via a reduction in KL-6, which is a biomarker of RA-ILD. Moreover, improvements in DLCO and diffusion coefficient percentages confirmed that baricitinib restored pulmonary function [122].

### 6.4. Upadacitinib

Upadacitinib is a second-generation JAK inhibitor selective for the JAK1 subtype. A patient with RA-ILD usually receives initial treatment with a csDMARD, mostly MTX monotherapy. However, when the patient is resistant to MTX, a combination of MTX with other csDMARDs or bDMARDs is recommended [123]. tsDMARDs, especially updacitinib, are recommended when a patient still does not show an adequate response to a combination of MTX and TNF inhibitor, which is a type of bDMARD [124]. A recent case report from Nishii et al. illustrated a case of a 69-year-old man with RA-ILD who previously did not respond to several treatments including MTX, TAC, ABA, and Cyc. However, upadacitinib therapy improved the patient’s pulmonary condition, which showed better DLCO and FVC. These results were consistent with a retrospective study of 43 patients with RA-ILD in Italy [125]. All three patients who were treated with upadacitinib exhibited stable high-resolution computed tomography (HRCT) results of pulmonary fibrosis. However, due to the insufficiency of clinical data, further investigations are anticipated to confirm the suitability of upadacitinib as an RA-ILD treatment.

## 7. Computational Methods for Further RA-ILD Drug Development

To prevent fibrosis and the progression of RA-ILD, and because there are limitations in applying IPF treatment to patients with RA-ILD, a novel drug for RA-ILD is necessary. However, considering that diseases with fibrotic characteristics are numerous and that their mechanisms share common molecules and genes, another solution for RA-ILD drug development could be drug repositioning or repurposing.

The first basis for drug repositioning is the phenotypic similarity between diseases, which has already been confirmed by studies that demonstrated shared molecules, genes, and signaling pathways between multiple organ fibrotic diseases [126]. Thus, computational drug repositioning is a favorable approach for identifying additional drugs with positive effects on RA-ILD. The drugs currently suggested for RA-ILD treatment also follow this notion because they were initially implemented for IPF or other CTD-ILDs. 

Although they were demonstrated only by in vitro assessments, certain clinical drugs for distinct diseases show potential for RA-ILD treatment. Cyclosporine, which is a well-known immunosuppressant, downregulates immune- and fibrosis-related genes, such as insulin-like growth factor binding protein 2 (IGFBP2), inhibitor of DNA binding protein 1 (ID1), and peroxisome proliferator-activated receptor gamma (PPARγ) [127]. Theophylline, which is a chronic obstructive pulmonary disease and asthma drug, is a pan-phosphodiesterase inhibitor that exhibits anti-inflammatory and anti-fibrotic effects by interfering with the TGF-β1 signaling pathway [128]. Fluvoxamine, which is a well-known antidepressant drug, showed equivalent anti-fibrotic effects with only one-third the dosage of pirfenidone [129]. These results suggested that multiple drugs that have already been approved for clinical safety can be used to control fibrosis.

For drug repositioning, potential targets that can effectively impede disease progression must be identified, as discussed in the following section. A computational platform for specifying the optimal targeting drug is introduced in the next section. In the forthcoming therapeutic development, the limitations of implementing RA-ILD in animal models and the great variety of patients with RA-ILD render the computational approach a more favorable option than traditional methods (Figure 3).

### 7.1. Molecular Targets Suggested for RA-ILD Fibrosis

Fibrosis is regulated by the transition, migration, and metabolic activity of fibroblasts and myofibroblasts. Excessive immune reactions in the lung tissue trigger cytokines and chemokines to be released and fibroblasts to be activated, which then differentiate into α-smooth-muscle-actin-expressing myofibroblasts. The invasive nature of pulmonary myofibroblasts stimulates cytoskeletal synthesis and upregulates the expression of contractile proteins. This contracts the lung tissue and substantial numbers of ECM proteins are produced [130,131,132,133].

Multiple studies suggested three molecular targets that can restrict fibroblast activity: TGF-β, CTGF, and MMP. These three targets inhibit and stimulate pulmonary fibrosis via multiple mechanisms and complex interactions. Genetic analysis demonstrated that they are deeply involved in multiple fibrotic diseases in different organs (eyes, heart, lungs, pancreas, and kidneys) [131,134].

TGF-β is a multifunctional cytokine secreted by bronchial structural cells or inflammatory cells that maintain tissue homeostasis. It causes the transition of epithelial and endothelial cells to mesenchymal cells (EMT and EndoMT), activates pericytes and smooth muscle cells related to blood vessels, and induces ECM deposition. Targeting TGF-β hinders signaling pathways that include SMAD 2/3, Wnt/β-catenin, MAPK, and PI3K [135,136,137]. The complex interplay between TGF-β and CTGF has been reported in pulmonary and renal fibrosis [131]. Different subtypes of MMPs have dual-purpose properties in pulmonary fibrosis. Dismantling specific collagen via MMP activation may delay fibrotic progression, whereas regulating MMP activity with a tissue inhibitor of metalloproteinase reduces inflammatory effects. Among these, MMP-1, -2, and -9 are marked with pulmonary fibrosis [134]. Targeting MMP-1 controls the local epithelial–mesenchymal interaction, MMP-2 plays a significant role in the EMT via the Wnt/β-catenin signaling pathway, and MMP-9 enhances recovery by decreasing IGFBP-3 expression [138,139,140].

Additional target molecules, including Syndecan-2, which promotes internalization and the degradation of the TGF-β receptor, are related to citrullinated peptides found in more than half of patients with RA and to the prevalence and exacerbation of ILD in patients with RA [141,142,143]. Additional targets can be found in other CTD-ILDs, including various autoimmune diseases, such as systemic lupus erythematosus, RA, Sjogren’s syndrome, polymyositis/dermatomyositis, systemic sclerosis (SSc), and mixed connective tissue disease invading the interstitial compartment of the lungs [144]. TGF-β, PDGF, and FGF are key growth factors that participate in the fibrotic process of most CTD-ILD diseases. In particular, the FGF and PDGF receptors and downstream signaling proteins, such as YAP/TAZ, are considered target molecules of SSc-associated lung fibrosis and RA-ILD [145].

### 7.2. Computational Approach for Drug Repurposing

Three primary components must be considered in the drug repositioning process: diseases, drugs, and their targets. By understanding their interconnections, researchers can gain insight into the efficacy of plausible treatments and identify potential side effects. Numerous data, such as chemical structures, transcriptional profiles, and phenotypic information, are already available. Thus, holistic data analysis is essential to determine the capacity of drug repositioning, an area in which in silico models have already stood out.

The computational drug repositioning of anti-fibrotic agents is currently in progress. Wu et al. [146] developed Dr AFC, which is a drug repositioning platform for anti-fibrotic characteristics, using two different models: structural and biological profile prediction models. As their names imply, each model uses chemical structures and compound-induced gene profiles to predict the anti-fibrotic efficacy of the given compound. Dr AFC demonstrated high predictability for the test sets and proposed several candidates as novel anti-fibrotic agents.

Similar to Dr AFC, similarity-based virtual screening using 2D and 3D structures of possible target molecules of RA-ILD and drug candidates can be performed to predict their affinity. Drugs showing a higher affinity for the target molecule can be selected and evaluated on the cell lines of SKG mouse models to further confirm their effectiveness in RA-ILD. The drug repositioning process can be streamlined by sorting drugs with high structural and biological probabilities using computational methods before proceeding to the next step.

Since diseases share molecular pathways and targets, multiple screening analyses on the molecular targets of RA-ILD have already been conducted, even though the studies’ primary goal was not RA-ILD. For instance, several possible drugs targeting PDGFRa were screened among 7200 drugs and natural compounds through structure-based virtual screening and molecular docking analysis. Specifically, three-dimensional quantitative structure–activity relationship (3D-QSAR) and absorption, distribution, metabolism, excretion, and toxicity analyses were performed [147]. Gimeno et al. [148] virtually screened 20 promising broad-spectrum MMP inhibitors from the Specs library and 18 of them showed their MMP inhibition ability through a bioactivity assay. Multiple targets can also be simultaneously considered in computational screening. Wang et al. [149] screened multi-targeting inhibitors of VEFGRs, VEGFRs, FGFRs, PDGFRs, and TGFβ1R as a potential IPF prodrug using virtual screening and machine learning models. Twenty compounds were selected from the first screening and went through a further verification process. Subsequently, two promising drug candidates were found with anti-fibrotic activity on mouse IPF models with low cellular toxicity. These studies demonstrate that computational screening gives valuable and reliable outcomes for further drug development processes. 

Such DTI prediction not only suggests a new drug but can also increase our knowledge about its potential side effects. For instance, Lounkine et al. [150] created a drug-target-adverse drug reaction network using a similarity ensemble approach, which predicts target binding based on chemical structure similarity with known ligands. New off-target and adverse events can be predicted using the network and can elucidate unknown mechanisms of adverse effects. One of the greatest obstacles to treating RA-ILD is the presence of multiple factors, such as joint, lung, and autoimmunity factors, that should be considered concurrently. If a drug benefits from one factor, but aggravates others, the drug is ineffective. Thus, we can predict off-target and adverse effects of the drugs in the early stages using these programs and filter out unsuitable options to accelerate the development process.

Computational analysis can help drug development to a further extent by identifying target molecules and signaling pathways that are involved in either the healing or adverse effects of drugs. A study by Zhu et al. [151] performed drug target analysis to understand the inexplicable healing effects of triptolide on CTD-ILDs. Using AutoDock Vina, they first discovered the top 10 target molecules of triptolide and further identified the signaling pathways. Another study aimed to understand the ILD that occurs in patients receiving TKI treatment. This treatment targeted EGFR, which is a key molecule of fibrosis. They analyzed the protein–protein interaction of EGFR and other proteins based on the String-bd database and discovered 37 molecules and 12 signaling pathways involved in EGFR-TKI-induced ILD [152]. 

Scholars are advancing in silico models with better algorithms and strategies to achieve higher prediction accuracy. In particular, the emergence and development of AI have changed the paradigm of drug discovery and drug repositioning. Traditional computational predictive approaches have relied on QSAR models utilizing machine learning techniques. These methods quickly calculated simple physicochemical properties from large amounts of data but were insufficient to predict complex properties, such as drug efficacy and adverse effects. Furthermore, these models face challenges with handling the variability of the dataset and their performance depends on the condition of the training datasets, such as their size or contamination with experimental errors [153]. However, with the development of AI, these issues are resolved as the computational screening techniques now encompass new models, such as deep learning (DL) and deep neural networks (DNN). For instance, DeepDTI, which was the first DL-based DTI prediction model developed by Wen et al. [154], outperformed other traditional machine learning techniques. In addition, the multitasking DNN-based model showed better predictability of drug toxicity compared with traditional QSAR models [155]. These advancements have also provided possibilities for precision medicine, which is suitable for RA-ILD because different biomarkers and pathogenic pathways are observed depending on the condition of the individual. Therefore, if these models can consider individual factors and help to prescribe customized drugs to each patient, this approach will be the optimal direction for future drug development.

## 8. Conclusions

RA-ILD is one of the most common lung manifestations of systemic inflammatory autoimmune disease, as roughly 10% of the patients with RA experience clinically significant RA-ILD [156]. Considering its high morbidity and mortality rates, the control of inflammation and fibrosis is of significant importance. This paper reviews the currently used agents for RA-ILD and describes a computational drug repositioning strategy that can benefit future drug developments for RA-ILD.

Nintedanib and pirfenidone are the most commonly used drugs for RA-ILD. The tyrosine kinase inhibitor, namely, nintedanib, controls the activity of fibrogenic cells (T cells, M2 macrophages, fibroblasts, and myofibroblasts) and decreases oxidative stress. Pirfenidone functions comprehensively via its anti-fibrotic, anti-inflammatory, and anti-oxidant effects. Multiple clinical trials were conducted using pirfenidone and nintedanib. They were shown to restore pulmonary function by slowing FVC decline and improving DLCO% and exercise capacity. Multiple studies demonstrated that immunosuppressants can stabilize or improve pulmonary function in patients with RA-ILD. However, the specific mechanism has not been clarified. Although more randomized controlled trials are required, regulating the autoimmune response can systematically affect multiple symptoms, including lung and arthritic manifestations. Similarly, several DMARDs exert auxiliary anti-fibrotic effects by controlling specific pathways involved in the progression of fibrogenesis. Drugs target signaling pathways, such as JAK/STAT, Hippo/Yap, Wnt/β-catenin, cGAS-STING, cAMP-PKA, and PI3K/AKT, to regulate cellular activity (differentiation, metabolic activity, or morphology), ECM homeostasis, and oxidative stress levels. While they differ only in the type of specific enzymes involved, these signaling pathways commonly regulate ECM homeostasis.

Although several drugs have been suggested and investigated, additional research in this area is required. This review summarizes three potent molecular targets that can impede fibrosis and a computational method that can determine the optimal drug to target.

Little is known about the pathogenesis of RA-ILD. Research conducted thus far has identified certain genetic backgrounds, environmental risk factors that trigger RA-ILD, and histologic evidence [8,157,158]. Thus, further research on the mechanism of RA-ILD, particularly focusing on the onset of the fibrotic stage and molecular correlations between joint inflammation and progressive fibrosis, is urgently needed. If the onset of the fibrotic stage can be clearly identified, preemptive actions, such as aggressive immunosuppressive treatment, can be administered in advance to inhibit subsequent fibrotic progression. Moreover, most drugs against RA are either predetermined to target joint inflammation or partially target fibrosis.

In contrast, anti-fibrotic agents inhibit joint inflammation. This implies that fibrosis and joint inflammation are interrelated in RA-ILD. In addition, deeper insights into the pathogenesis can resolve adverse events that frequently occur with current therapeutic treatments. Ultimately, the pathogenesis of RA-ILD must be elucidated to develop new drugs and reverse adverse effects.

## Figures and Tables

**Figure 1 ijms-25-02682-f001:**
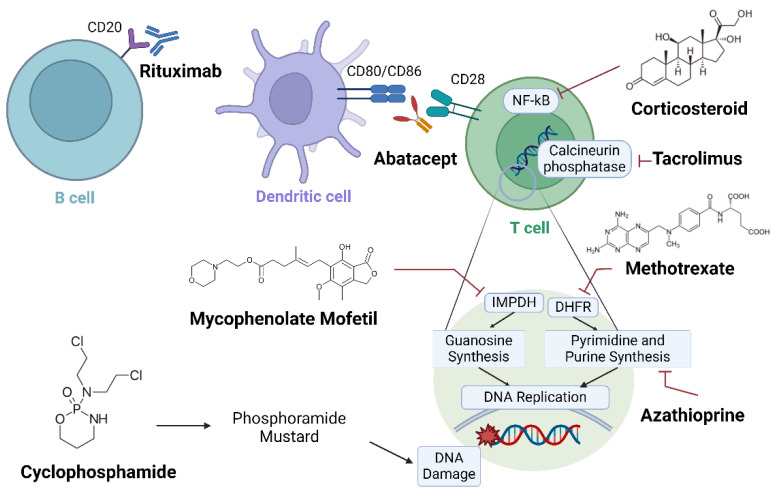
Schematic diagram in which immunosuppressants target rheumatoid arthritis-interstitial lung disease (RA-ILD). Mechanisms of several immunosuppressants against RA-ILD are presented. Therapeutic antibodies, such as rituximab and abatacept, immobilize immune cells, and anti-metabolites cause DNA damage. Multiple clinical trials have been conducted to evaluate their efficacy against RA-ILD. Created with BioRender.com (accessed on 22 Feburary 2024).

**Figure 2 ijms-25-02682-f002:**
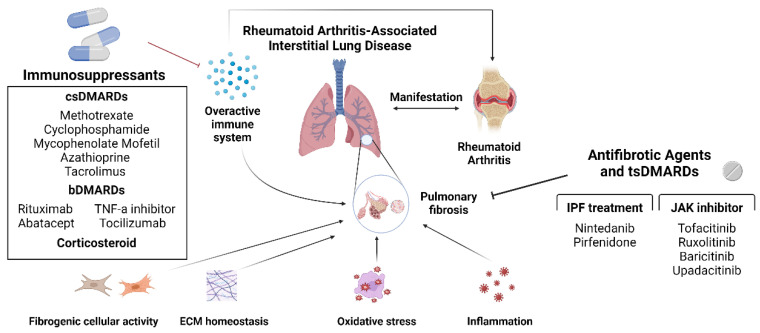
Schematic diagram of the currently used drugs against RA-ILD. Immunosuppressants and anti-fibrotic agents control joint inflammation and pulmonary fibrosis to impede the fibrotic process of RA-ILD. Pulmonary fibrosis can be controlled via four mechanisms: diminishing fibrogenic cellular activity, restoring ECM homeostasis, decreasing oxidative stress, and controlling inflammation. Clinical trial results demonstrated that drugs for IPF treatment, nintedanib, and pirfenidone have anti-fibrotic effects against RA-ILD. Four JAK inhibitors, tofacitinib, ruxolitinib, baricitinib, and upadacitinib are targeted synthetic DMRARDs and not pure anti-fibrotics, but studies suggest their anti-fibrotic effects to a certain point. Created with BioRender.com (accessed on 22 Feburary 2024).

**Figure 3 ijms-25-02682-f003:**
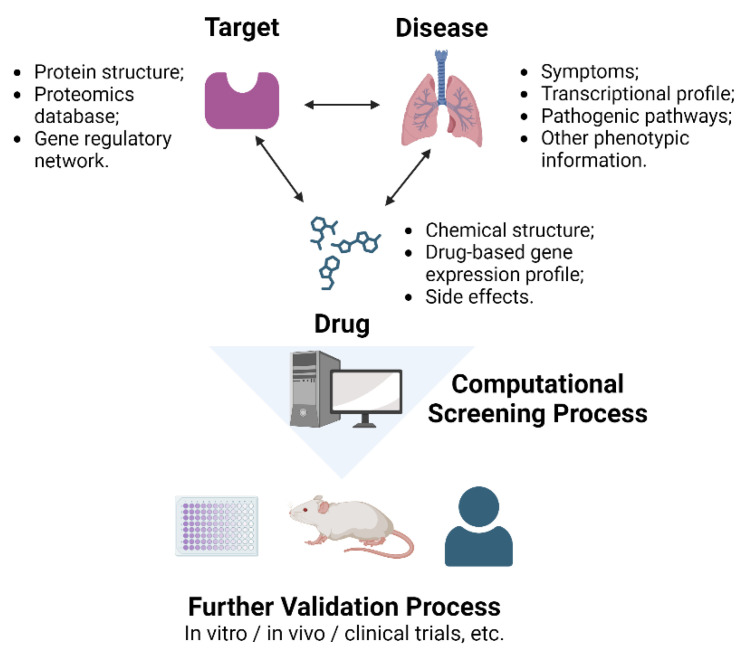
Schematic diagram of using features of the disease to identify potential targets and applying computational screening processes to screen potent drugs for further validation processes. Drug repositioning via computational screening can be an efficient process for further RA-ILD drug development. Pathologic features of RA-ILD can be acquired through its symptoms, transcriptional profile, pathogenic pathways, or other phenotypic information. Through this process, potential targets for drugs can be identified, and three potential targets were suggested: TGF-β, CTGF, and MMP. Computational platforms can embody the 2D and 3D structures of a specific target and calculate its interaction with multiple candidate drugs and screen potent drugs. Moreover, possible adverse effects of the drugs can also be identified, which will be helpful in further in vivo and in vitro or clinical trials. Created with BioRender.com (accessed on 22 Feburary 2024).

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
