# Peer review of "Potential Rheumatoid Arthritis-Associated Interstitial Lung Disease Treatment and Computational Approach for Future Drug Development"

_ijms, 2024, doi:10.3390/ijms25052682_

Round 1
Reviewer 1 Report
Comments and Suggestions for Authors
To the Authors:
I congratulate you on writing this review, I’ve a comment:
Comment 1. Introduction, page 2, row 70: I believe the authors wrongly stated that both pirfenidone and nintedanib are used as off-label treatments for RA-ILD. Nintedanib, as correctly reported later in the paper received the FDA approval for chronic fibrosing ILD with a progressive phenotype, regard- 119 less of the underlying cause. I’d correct this sentence at page 2.
Comments on the Quality of English LanguageN/A
Author Response
Dear Editor-in Chief,
We would like to appreciate the reviewers for their every constructive advice and critiques, which have assisted in improving our review article, “Potential Rheumatoid Arthritis-associated Interstitial Lung Disease Treatment and Computational Approach for Future Drug Development”. We have now addressed all of the reviewer's comments and have provided point-by-point responses in yellow to comments in the following paragraphs.
We would like to thank the reviewers again for all their comments to improve the manuscript.
Reviewer 1’s comments
Introduction, page 2, row 70: I believe the authors wrongly stated that both pirfenidone and nintedanib are used as off-label treatments for RA-ILD. Nintedanib, as correctly reported later in the paper received the FDA approval for chronic fibrosing ILD with a progressive phenotype, regard- 119 less of the underlying cause. I’d correct this sentence at page 2.
Response: Thank you for the valuable comment. We revised the sentence about Nintedanib to clarify the description on page 2, line 70. “In case of nintedanib, the treatment recently received FDA approval as the first treatment for chronic fibrosing ILD with progressive trait, supporting its use in RA-ILD”
Reviewer 2 Report
Comments and Suggestions for Authors
In this narrative review, the Authors aimed to summarize current RA-ILD treatment approaches and provide an overview on computational approach for future drug development. In the first part of the review the current drug used for RA-ILD treatment is presented, starting from anti-fibrotics to b and ts-DMARDs. In the second part, the Authors speculated that the use of a computational screening process can be an efficient way for further RA-ILD treatment discovery, discussing the most promising molecular targets, such as TGF-beta, CTGF and MMP. The pro-fibrotic role of these molecules is well known and attempts to design an active molecule have been made, especially for TGF-beta, but were not successful.
Although interesting, the manuscript needs some revision.
1. I understand the focus on RA-ILD but similar studies and reviews in different ILDs have been published already and should be taken into account (i.e. by presenting supposed differences or similarities) or at least cited. In particular, the paragraph 5.2 should be more precise in presenting the available results in virtual screening analysis in the molecules that have been cited earlier in the text. for example, for PDGF see ref. Int J Mol Sci. 2023 Jun 1;24(11):9623. doi:10.3390/ijms24119623.
Additionally, the role of AI in the prediction of interaction between drugs and targets has not been explored at all.
2. Page 2. line 67-70. The use of nintedanib in RA-ILD with progressive features is not off-label anymore. The importance of progressive disease (PPF as per ATS 2022 definitino) in RA-ILD should be discussed in the introductive paragraph.
3. Page 4, line 193. RELIEF trial is misspelled.
4. PAge 6, line 297. The reference for the RECITAL trial [29] is wrong. Please, check all references in the text.
5. figure 2. JAK inhibitors are not pure anti-fibrotics, I would tag them as anti-inflammatory or targeted syntethic DMARDs.
6. Page 13, line 556. The prevalence of RA is around 1% but that of RA-ILD is much lower. Please correct.
Comments on the Quality of English Language
English is fine but spell and grammar checks are needed.
Author Response
Dear Editor-in Chief,
We would like to appreciate the reviewers for their every constructive advice and critiques, which have assisted in improving our review article, “Potential Rheumatoid Arthritis-associated Interstitial Lung Disease Treatment and Computational Approach for Future Drug Development”. We have now addressed all of the reviewer's comments and have provided point-by-point responses in yellow to comments in the following paragraphs.
We would like to thank the reviewers again for all their comments to improve the manuscript.
Reviewer 2’s comments
I understand the focus on RA-ILD but similar studies and reviews in different ILDs have been published already and should be taken into account (i.e. by presenting supposed differences or similarities) or at least cited. In particular, the paragraph 5.2 should be more precise in presenting the available results in virtual screening analysis in the molecules that have been cited earlier in the text. for example, for PDGF see ref. Int J Mol Sci. 2023 Jun 1;24(11):9623. doi:10.3390/ijms24119623.
Additionally, the role of AI in the prediction of interaction between drugs and targets has not been explored at all.
Response: Thank you for the valuable comment and the example article. We provided one example of virtual screening analysis related to fibrotic diseases, Dr AFC. The reason was that we could not find suitable virtual screening studies that specifically targeted these molecules in “fibrotic” diseases. However, based on your example article, we could provide other examples that analyzed the target molecules of RA-ILD, even though their studies’ primary goal was not RA-ILD. It was great help to improve the manuscript. According to the reviewer comments, we added an additional paragraph on page 13 line 613- 629 and included three studies.
Moreover, during the revision, we perceived that computational analysis for drug development can benefit explaining the mechanisms behind the healing or adverse effects of drugs which were previously not understood. We therefore added two examples related to the notion on page 14 line 642- 651.
Lastly, on your critical comment of the role of AI in the DTI prediction, we found that the AI can greatly help overcome errors and variabilities of the datasets in the process. We’ve included to our manuscript by stating how the AI can enhance the conventional QSAR models on page 14 line 654- 667. Thank you again for your helpful comments.
- Page 2. line 67-70. The use of nintedanib in RA-ILD with progressive features is not off-label anymore. The importance of progressive disease (PPF as per ATS 2022 definition) in RA-ILD should be discussed in the introductive paragraph.
Response: Thank you for the valuable comment. We’ve added a statement to clarify the description of Nintedanib on page 2 line 70. “In case of nintedanib, the treatment recently received FDA approval as the first treatment for chronic fibrosing ILD with progressive trait, supporting its use in RA-ILD.”
- Page 4, line 193. RELIEF trial is misspelled.
Response: Thank you for the valuable comment. We made changes to the spelling on page 4 line 195.
- Page 6, line 297. The reference for the RECITAL trial [29] is wrong. Please, check all references in the text.
Response: Thank you for the valuable comment. We made changes to the reference on page 7 line 344, and checked all the other references in the text.
- figure 2. JAK inhibitors are not pure anti-fibrotics, I would tag them as anti-inflammatory or targeted synthetic DMARDs.
Response: Thank you for the valuable comment. We made changes to figure 2 and classified JAK inhibitors as targeted synthetic DMARDs. We revised the line on page 11 line 501-503 and stated that JAK inhibitors are synthetic DMARDs and not pure anti-fibrotics, but that they have shown anti-fibrotic effects to a certain point in clinical trials.
- Page 13, line 556. The prevalence of RA is around 1% but that of RA-ILD is much lower. Please correct.
Response: Thank you for the critical comment. To correct the statement, we attempted to find the global prevalence of RA-ILD but failed to find credible information. We therefore changed the statement to the following on page 15 line 685-687 : “RA-ILD is one of the most common lung manifestations of systemic inflammatory autoimmune disease, a roughly 10% of the patients with RA experience clinically significant RA-ILD”.
Round 2
Reviewer 2 Report
Comments and Suggestions for Authors
I have no further comments
Comments on the Quality of English LanguageEnglish is fine but minor spell checks are needed